# Laboratory evaluation of the rapid diagnostic tests for the detection of *Vibrio cholerae* O1 using diarrheal samples

**Goutam Chowdhury**[1], **Tarosi Senapati**[2], **Bhabatosh Das**[2], **Asha Kamath**[3], **Debottam Pal**[4], **Puja Bose**[1], **Arundhati Deb**[2], **Sangita Paul**[1], **Asish K. Mukhopadhyay**[1], **Shanta Dutta**[1], **Thandavarayan Ramamurthy**[1,2]*

1 Division of Bacteriology, ICMR-National Institute of Cholera and Enteric Diseases, Kolkata, India, 2 Molecular Genetics Laboratory, Infection and Immunology Division, Translational Health Science Institute of Technology, Faridadad, India, 3 Department of Data Science, Prasanna School of Public Health, Kasturba Medical College, Manipal, India, 4 Division of Epidemiology, ICMR-National Institute of Cholera and Enteric Diseases, Kolkata, India

☯ These authors contributed equally to this work.
* ramamurthy.t@icmr.gov.in

## Abstract

### Background

Cholera, an acute diarrheal disease is a major public health problem in many developing countries. Several rapid diagnostic tests (RDT) are available for the detection of cholera, but their efficacies are not compared in an endemic setting. In this study, we have compared the specificity and sensitivity of three RDT kits for the detection of *Vibrio cholerae* O1 and compared their efficiency with culture and polymerase chain reaction (PCR) methods.

### Methods

Five hundred six diarrheal stool samples collected from patients from two different hospitals in Kolkata, India were tested using SD Bioline Cholera, SMART-II Cholera O1 and Crystal-VC RDT kits. All the stool samples were screened for the presence of *V. cholerae* by direct and enrichment culture methods. Stool DNA-based PCR assay was made to target the cholera toxin (*ctxAB*) and O1 somatic antigen (*rfb*) encoding genes. Statistical evaluation of the RDTs has been made using STATA software with stool culture and PCR results as the gold standards. The Bayesian latent class model (LCM) was used to evaluate the diagnostic tests in the absence of the gold standard.

### Results

Involving culture technique as gold standard, the sensitivity and specificity of the cholera RDT kits in the direct testing of stools was highest with SAMRT-II (86.1%) and SD-Cholera (94.4%), respectively. The DNA based PCR assays gave very high sensitivity (98.4%) but the specificity was comparatively low (75.3%). After enrichment, the high sensitivity and specificity was detected with SAMRT-II (78.8%) and SD-Cholera (99.1%), respectively. Considering PCR as the gold standard, the sensitivity and specificity of the RDTs remained

**Data Availability Statement:** All relevant data are within the manuscript and its Supporting Information files.

**Funding:** The work was supported by the Bill & Melinda Gates Foundation (Investment ID-OPP1126286). The funders had no role in the study design, data collection and analysis, decision to publish, or preparation of the manuscript.

**Competing interests:** The authors have declared that no competing interests exist.

between 52.3–58.2% and 92.3–96.8%, respectively. In the LCM, the sensitivity of direct and enrichment testing was high in SAMRT-II (88% and 92%, respectively), but the specificity was high in SD cholera for both the methods (97% and 100%, respectively). The sensitivity/specificity of RDTs and direct culture have also been analyzed considering the age, gender and diarrheal disease severity of the patients.

## Conclusion

Overall, the performance of the RDT kits remained almost similar in terms of specificity and sensitivity. Performance of PCR was superior to the antibody-based RDTs. The RTDs are very useful in identifying cholera cases during outbreak/epidemic situations and for making them as a point-of-care (POC) testing tool needs more improvement.

## Author summary

Cholera is caused by toxigenic *Vibrio cholerae*, which induces massive fluid accumulation in the host's gut and secretory diarrhea. Cholera deaths can be prevented by timely diagnosis and early treatment of the patients using rehydration therapy. Outbreaks of cholera are often reported in several countries due to poor quality of drinking water and lack of sanitation. Early diagnosis of cholera outbreaks is highly useful for the enforcement of control measures. In many cholera endemic countries, laboratory resources in detecting the cholera cases are limited. Even though the conventional culture methods of the isolation and identification *V. cholerae* are useful for cholera diagnosis, its sensitivity is not superior compared to antibody and DNA-based techniques. Several antibody-based cholera rapid diagnostic kits (RTDs) are designed for use as a point-of-care (POC) device or field conditions. Using the diarrheal stool samples, we compared the performance of three cholera RDTs with bacterial culture and PCR assays. Applying culture and PCR results as the gold standards and also in the absence of a gold standard, appropriate statistical analysis has been made for diagnostic test evaluations. We have also considered the presence of other pathogens in the stools and clinical characteristics of the patients in the analysis. Though the cholera RDT kits highly useful for the detection of *V. cholerae* O1, even in the presence of other pathogens in the stools, they cannot be considered as a POC tool due to lack of required specificity.

## Introduction

Cholera is a major public health problem in many developing countries. In 2017, 34 countries reported more than 490,000 cholera cases and 2900 deaths to WHO [1]. Significant epidemiological events in the history of cholera include Latin American epidemic after 100 years [2], genesis of *Vibrio cholerae* O139 in the Indian subcontinent [3] and the emergence and spread of *V. cholerae* hybrid El Tor strains [4]. In October 2010, the Haitian cholera epidemic affected more than 8,00,000 individuals with 9,000 deaths [5]. In Yemen, more than one million cholera cases and 2300 deaths were reported between 2016 and 2018 that represent an overall attack rate of 3·7%, which is one of the largest epidemics in Asia [6].

The causative agent of cholera is a Gram-negative curved bacterium *Vibrio cholerae*. Cholera toxin (CT) produced by this pathogen is the principal virulence factor associated with the

disease. CT comprise A and B subunits, which are encoded by the *ctxA* and *ctxB* genes, respectively. Both the genes are part of a filamentous ssDNA bacteriophage CTXΦ, which is integrated into the *dif* loci of the chromosome of *V. cholerae* [7]. The CT-A subunit is responsible for the disease phenotype, while the B subunit transports subunit A to target cells by catalyzing adenosine diphosphate (ADP)-ribosylation, leading to stimulation of adenylate cyclase and increase intracellular cyclic adenosine monophosphate (cAMP) [8]. Rise in intracellular cAMP results in reduced sodium uptake and increased chloride outflow, triggering the profuse water secretion in the form of acute diarrhea, which may become fatal if untreated [9].

*V. cholerae* is classified into more than 200 somatic O antigen serogroups [10,11]. The O1 serogroup is differentiated into two biotypes, classical and El Tor, both comprise Ogawa and Inaba serotypes. The classical biotype was presumed to be involved in first six cholera pandemics and the El Tor biotype is associated with the ongoing seventh pandemic [12]. The other toxigenic *V. cholerae* serogroup O139, synonym Bengal, has emerged in the Indian subcontinent during 1992 and spread to other Asian countries [3]. Both the O1 and O139 serogroups are known to cause epidemic cholera.

In clinical settings, acute cholera cases are recognized based on the characteristic clinical symptoms typified by rice watery diarrhea with or without vomiting and severe dehydration. Mortality due to cholera can be prevented by timely detection of the disease and replacement of fluid loss by rehydrating the affected patients. Since several other pathogens can induce acute diarrhea, conventional methods are being followed in the identification of the causative agent. Cholera diagnosis has been made by isolation and identification of *V. cholerae* from stool specimens. These culture based methods might take two or more days and also demands good laboratory infrastructure with skilled staff. Delayed detection of cholera outbreaks may have several consequences including poor public health actions, spread of the disease and increase in morbidity and mortality rates. Rapid diagnosis helps in the establishment proper care at the early stage of infection and timely implementation of interventions in all settings.

Cholera rapid diagnostic test (RDT) represents promising tools in the early detection of *V. cholerae* O1/O139 directly from the stool specimens even in remote areas where laboratory resources are poor [13]. This technique requires no special laboratory skills for the detection of cholera cases [14]. Considering its prominence, RDT has been included in the WHO's cholera investigation (https://www.who.int/cholera/kit/cholera-kit-item-list.pdf?ua=1). One of widely used cholera RDTs is Crystal-VC (Arkray Health Care Pvt Ltd, Surat, India), which is a vertical flow dipstick kit. Almost all the cholera RDT kits are based on the detection of antigen specific for the lipopolysaccharides (LPS) of *V. cholerae* O1 and O139 serogroups by monoclonal antibodies that works following the principle of immunochromatography [15]. In several studies, it was shown that the sensitivity and specificity of cholera RDTs vary and hence used as an epidemiological tool rather than a diagnostic kit [16,17]. In addition, RDT has been used in a cholera immunization campaign to identify vaccinated individuals [18]. Several cholera RDT kits are available, but their performance is not validated with a sizeable number of samples. In addition to the culture and antibody-based RDTs, several polymerase chain reaction (PCR) techniques have been used for the rapid detection of virulence genes of *V. cholerae* [19]. PCR assay is generally faster and more sensitive than the culture methods and hence identification of *V. cholerae* in stool by culture or PCR is considered the gold standard for cholera diagnosis [20–22]. However, these assays demand suitable laboratory infrastructure, expensive equipment and skilled staff, which would not exist in remote areas where outbreaks often occur.

In this study, we have considered three different cholera RDT kits, not only evaluate their performances, but also compare the results with culture and PCR based techniques. The primary aim of the study was to determine whether the overall diagnostic performance of RDTs was equivalent to, or better than other diagnostic methods. In addition, we have made an

attempt to provide estimates of the sensitivity/specificity stratified by disease severity, age and gender. We have also tested the detection limit of the cholera RDT kits and duration of *V. cholerae* O1 viability in Cary-Blair medium to check their effective use.

## Materials and methods

### Ethics statement

The Ethical Review Committee of the National Institute of Cholera and Enteric Diseases (NICED), Kolkata has approved this study. Written informed consent was obtained from the study participants or from parents/guardians in the case of minors. Privacy and confidentiality of the data collected from participants was ensured during and after the study.

### Sample collection

Before the administration of antibiotics, stool specimens were collected from the hospitalized diarrheal patients in the Infectious Diseases Hospital (IDH) and children treated for diarrhea as outpatients in the B. C. Roy Children Hospital (BCH), Kolkata. A diarrheal infection is defined as a patient passes 3 or more loose or liquid stools in last 24 hrs or less than 3 loose/liquid stools associated with dehydration; or at least one bloody loose stool in last 24 hrs. Clinical symptoms of diarrheal patients included loose/watery stools, with or without dehydration, abdominal cramps, vomiting and fever. Dysentery patients had frequent passage of stool with blood/mucus and mild to severe abdominal pain. Stool specimens were collected in sterile wide-mouthed containers and transported within 2 hrs to the laboratory of NICED at ambient temperature. Sampling was made during two consecutive peak cholera seasons, i.e., from August-December 2016 and July-November 2017. Samples were not considered in this study if the patients had the history of using antibiotics before visiting the hospital, as it may affect with the culture results and eliminate/reduce the number of the pathogens in the stools.

The sample size for the study was based on Buderer's formula for sensitivity and specificity of diagnostic health sciences. Results from a previous hospital study [23] reported 26% isolation of *V. cholerae* O1 from diarrhea patients of all age groups. Considering 90% specificity for the test the required sample size would be 260, for 95% confidence level and 6% absolute precision. Similarly, with 80% sensitivity of the test the required sample size would be 463. Expecting variation in the isolation proportion of the organism, we included approximately 10% more sample and finally included 506 samples in this study.

### Cholera RDTs

For screening, we used three cholera RDT kits, namely, the SD Bioline Cholera Ag O1/O139 (Standard Diagnosis, S. Korea), SMART-II Cholera O1 (New Horizon, USA) and Crystal-VC (Arkray Health Care Pvt Ltd, Surat, India). Of these, SD Bioline Cholera and Crystal-VC can detect O1 and O139 serogroups of *V. cholerae*. Five drops (~200 µl) of liquid stool were added into the sample processing vial and mixed gently with the diluent supplied along with the respective kit. *V. cholerae* allowed to grow in the alkaline peptone water (APW, pH 8.0) for 4 hrs from the stool specimens were also tested using the RDT kits. For the Crystal-VC test, four drops of the processed sample was placed in a test tube and the test strip was dipped into the tube for vertical flow. The results were interpreted according to the manufacturer's protocol. For the other two lateral flow kits, the diluted stool was dispensed directly onto the sample well of the test cassette and the results were interpreted as per the recommendation of manufacturers.

## Culture technique

For the isolation of *V. cholerae*, stool specimens were directly inoculated on thiosulphate citrate bile-salts sucrose agar (TCBS, Eiken, Tokyo, Japan) plates, followed by overnight incubation at 37°C. Simultaneously, few drops of stools were inoculated in APW for 4 hrs. APW enriched culture was screened for *V. cholerae* using TCBS agar. Typical sucrose-positive *V. cholerae* isolates were sub-cultured on Luria Bertani agar (LB, Difco, Sparks, MD, USA) and serologically tested using commercially available *V. cholerae* O1 poly and Ogawa and Inaba monovalent antisera (Denka-Seiken, Tokyo, Japan). Stool specimens were also tested for other enteric bacterial, viral and protozoan pathogens following the methods outlined in our previous study [23].

## PCR

Two hundred μl of watery stool was used for DNA extraction using QIAamp Fast DNA stool mini kit (Qiagen, Hilden, Germany) following the manufacturer's instructions. At the Translational Health Science and Technology Institute, Faridabad, PCR assay was performed with stool DNA, targeting the CT gene (*ctxAB*) as well as the *rfb* region that encodes somatic antigen of the O1 serogroup. Simplex PCR assay was performed on an Eppendorf Mastercycler instrument (Eppendorf, Germany). *V. cholerae* O1-*rfb* specific primers: O1-F(5′-TCTATGTG CTGCGATTGGTG-3′), O1-R(5′-CCCCGAAAACCTAATGTGAG-3′) and cholera toxin (*ctxAB*) gene primers: *ctx*-F (5′- CAATATCAGATTGATAGCCTGA-3′), *ctx*-R (5′-ACTA ATTGCGGCAATCGCATG-3′) were used to amplify O1 *rfb* (amplicon size 650 bp), and *ctx* (amplicon size 413 bp) genes, at an annealing temperatures of 46°C and 49°C for 20 sec, respectively. PCR products were visualized on a 1% agarose gel using Gel imager (AlphaImager HP, San Jose, CA, USA) after staining with ethidium bromide (0.5 μg/ml). DNA extracted from *V. cholerae* O1 strain N16961was used as positive control. This study was made by fully blinding trained technicians while performing the culture of stools, RDTs and PCRs.

## Detection limit of the cholera RDT kits

Overnight culture of the standard *V. cholerae* O1 strain N16961was used as a source culture in LB broth (Difco) that was grown to log phase at 37°C for 4 hrs. Serial dilutions (from $10^1$ to $10^6$) of this culture were made using sterile phosphate buffer saline (PBS, pH 7.0). All the three RDT kits were tested with an aliquot from each dilution. To count the bacterial colonies, 100 μl of each dilution was also plated on LB agar (Difco) plates and incubated overnight at 37°C.

## Duration of *V. cholerae* O1 viability in Cary-Blair medium

Swabs were wetted with several dilutions of *V. cholerae* O1 culture and kept in Cary-Blair transport medium (Difco) at ambient temperature. The duration of viability and performance of cholera RDTs were assayed at regular intervals till 18 days using APW enriched cultures.

## Data analysis

The clinical and laboratory data were checked manually and entered into pre-designed data entry proforma developed in visual basic with inbuilt entry validation checking facilitated program in structure query language (SQL) server by the dual entry method. Data was randomly checked and matched to derive consistency and validity for analysis.

For analysis purpose, a true cholera case was confirmed by culture positive for *V. cholerae* O1. A true negative case was delineated with culture negative for the target pathogen. The

sensitivity, specificity, positive predictive value, negative predictive value and accuracy were estimated for both direct culture testing and testing after 4 hrs of enrichment in APW followed by growth in TCBS agar and serological confirmation and also by comparing the three RDT results to the culture reference standard.

The primary endpoint is the valuation of the RDT using stool culture results, i.e. both by direct and enrichment techniques for the isolation of *V. cholerae* O1, as the gold standard for comparison. In addition, a separate analysis for the performance of the RDTs using PCR as the gold standard was also made. Sensitivity was defined as the probability that patients with culture-confirmed cholera had a positive RDT. Specificity was identified as the probability that patients with no culture-confirmed cholera had a negative RDT. The positive predictive value (PPV) was the probability that patients with a positive RDT had *V. cholerae* O1 isolated from stool culture. The negative predictive value (NPV) was the probability that patients with a negative RDT had no *V. cholerae* isolated from a stool culture. Statistical analyses were performed using the software for statistics and data science (STATA version 13, Stata Corp, Texas, USA). Sensitivity and specificity were verified based on the comparison of RDT results with the culture test as well as PCR assay and presented as percentages. For better predictions, a 95% Clopper-Pearson confidence intervals (CIs) were also estimated.

The Bayesian latent class model (LCM) combines the established hypotheses on test characteristics with actual results to evaluate the performance of each assay. The Bayesian-LCM was used in evaluating diagnostic tests in the absence of a "gold standard" test under the assumptions of a two-test, two population latent class model, namely, (i) each population prevalence should be different when multiple populations are being compared; (ii) the sensitivity ($S_e$) and specificity ($S_p$) of the test are the same across test populations; and (iii) the tests are conditionally independent [24]. The analysis was performed using five tests and one population [25,26]. Assuming a multinomial distribution for the counts of the different combinations of the test results of the five tests, and a Dirichlet prior, the model parameters were estimated under conditional independence [25,26]. All models were applied using the R statistical software environment using the package BayesLCA. Using Gibbs sampling, posterior inference was performed and the estimates with the Bayesian 95% credibility intervals were reported.

The accuracy was defined as the percentage of correctly classified instances (TP + TN)/ (TP + TN + FP + FN), where TP, FN, FP and TN represent the number of true positives, false negatives, false positives and true negatives, respectively. Using the specificity and positive and negative predictive values, a probabilistic clinical utility index (CUI) was made for an application of multiattribute utility that focuses on clinical attributes.

## Results

The clinical characteristics of patients included in this study are shown in **Table 1**. Of the 506 stool specimens tested, 243 (48%) were negative for all the assays used in this study and 91 (18%) were positive in all the tested assays. The rest of the 172 (34%) samples yielded positive result at least in any one of the assays. Among the total of 506 stool samples, *V. cholerae* O1 was isolated directly from 129 (25.5%) samples and 156 (30.8%) by enrichment culture. SMART-II test kit gave highest positive results with direct stool (29.6%; 150/506) and enriched sample assays (28.1%; 142/506). The performance of SD cholera and Crystal VC was nearly the same in both by direct and enrichment samples. The stool DNA based PCR assay gave the highest number of positive samples (43.5%, 220/506).

The results of conventional approach with direct culture technique using the stool samples as gold standard are shown in **Table 2**. The sensitivity of the cholera RDT kits in the direct testing was high in SMART-II (86.1%), followed by Crystal-VC (82.6%). The specificity was

**Table 1. Clinical characteristics of patients included in this study.**

| Clinical features | | No. of patients (%) *n* = 506 |
|---|---|---|
| Age | <5 years | 73 (14.4) |
| | >5 years | 433 (85.6) |
| Sex | Male | 256 (50.6) |
| | Female | 250 (49.4) |
| Types of diarrhea | Watery | 376 (74.3) |
| | Loose | 116 (22.9) |
| | Bloody | 3 (0.6) |
| | Mucoid | 1 (0.2) |
| | Bloody-mucoid | 10 (2.0) |
| Fever | Yes | 212 (41.9) |
| | No | 294 (58.1) |
| Abdominal pain | Yes | 209 (41.3) |
| | No | 297 (58.7) |
| Dehydration | Severe | 91 (18.0) |
| | Some | 415 (82.0) |

high in SD-Cholera (94.4%) followed by Crystal-VC (93.6%). The accuracy was high in SD-Cholera and Crystal-VC (>90%). Overall, the DNA based *ctx* and O1*rfb* PCR assays had a very high sensitivity (98.4%) but the specificity (75.3%) and accuracy (81.2%) were comparatively low. The CUI with the positive predictive value was good with SD-Cholera and Crystal-VC with the attribute weight of more than 0.64. However, the CUI was fair in the case of SMART-II and PCRs with the attribute weight of 0.637 and 0.568, respectively. The CUI with the negative predictive value was excellent in all the RDTs with the attribute weight of more than the cutoff value of 0.81.

The performance results of RDTs after enrichment culture as the gold standard is shown in **Table 3**. The sensitivity of the kits was >78% for SAMRT-II and Crystal-VC and the specificity was high in SD-Cholera (99.1%). Accuracy was high in Crystal-VC (92.1%), followed by

**Table 2. Assay results of RDTs, direct culture and PCR (Culture technique as gold standard).**

| Assay | | Culture | | Sens. | Spec. | PPV | NPV | Accuracy | CUI+* | CUI-* | P-value# |
|---|---|---|---|---|---|---|---|---|---|---|---|
| | | Positive | Negative | | | | | | | | |
| SD Cholera | Positive | 105 | 21 | 81.4% | 94.4% | 83.3% | 93.7% | 91.1% | 0.678 | 0.884 | <0.0001 |
| | Negative | 24 | 356 | (73.6–87.7) | (91.6–96.5) | (75.7–89.4) | (90.8–95.9) | (88.3–93.4) | | | |
| SMART—II | Positive | 111 | 39 | 86.1% | 89.7% | 74.0% | 94.9% | 88.7% | 0.637 | 0.851 | <0.0001 |
| | Negative | 18 | 338 | (78.9–91.5) | (86.1–92.5) | (66.2–80.8) | (92.1–97) | (85.7–91.4) | | | |
| Crystal VC | Positive | 107 | 24 | 82.6% | 93.6% | 81.7% | 94.1% | 90.9% | 0.675 | 0.881 | <0.0001 |
| | Negative | 22 | 353 | (75.3–89) | (90.7–95.9) | (74–87.9) | (91.2–96.3) | (88.1–93.3) | | | |
| PCR *ctx* | Positive | 127 | 93 | 98.4% | 75.3% | 57.7% | 99.3% | 81.2% | 0.568 | 0.748 | <0.0001 |
| | Negative | 2 | 284 | (94.5–99.8) | (70.6–79.6) | (50.9–64.3) | (97.5–99.9) | (77.6–84.5) | | | |
| PCR *rfb* | Positive | 127 | 93 | 98.4% | 75.3% | 57.7% | 99.3% | 81.2% | 0.568 | 0.748 | <0.0001 |
| | Negative | 2 | 284 | (94.5–99.8) | (70.6–79.6) | (50.9–64.3) | (97.5–99.9) | (77.6–84.5) | | | |

Number in the parentheses indicate the range

*CUI; Clinical utility index of the positive and negative test. >0.81 excellent utility; 0.64–0.80 good utility

0.49–0.63 fair utility; 0.36–0.48; poor utility and <0.36 very poor utility [34]

#Fisher's exact test.

**Table 3. Assay results of RDTs and enrichment culture (Culture technique as gold standard).**

| Assay | | Enriched Culture | | Sens. | Spec. | PPV | NPV | Accuracy | CUI+* | CUI-* | p-value# |
|---|---|---|---|---|---|---|---|---|---|---|---|
| | | Positive | Negative | | | | | | | | |
| SD Cholera | Positive | 111 | 3 | 71.2% (63.4–78.1) | 99.1% (97.5–99.8) | 97.4% (92.5–99.5) | 88.5% (84.9–91.5) | 90.5% (87.6–92.9) | 0.693 | 0.877 | <0.0001 |
| | Negative | 45 | 347 | | | | | | | | |
| SMART-II | Positive | 123 | 19 | 78.8% (71.6–85) | 94.6% (91.7–96.7) | 86.6% (79.9–91.7) | 90.9% (87.5–93.7) | 89.7% (86.7–92.2) | 0.682 | 0.854 | <0.0001 |
| | Negative | 33 | 331 | | | | | | | | |
| Crystal VC | Positive | 122 | 6 | 78.2% (70.9–84.4) | 98.3% (96.3–99.4) | 95.3% (90.1–98.3) | 91.0% (87.6–93.7) | 92.1% (89.4–94.3) | 0.745 | 0.894 | <0.0001 |
| | Negative | 34 | 344 | | | | | | | | |

Number in the parentheses indicate the range

*CUI; Clinical utility index of the positive and negative test. >0.81 excellent utility; 0.64–0.80 good utility

0.49–0.63 fair utility; 0.36–0.48; poor utility and <0.36 very poor utility [34].

#Fisher's exact test.

SD-Cholera (90.5%). The CUI with the positive predictive value was good in all the RDTs with the attribute weight of more than the cutoff value (0.64). The CUI with the negative predictive value was excellent in SD-Cholera and Crystal-VC kits with the attribute weight of more than the cutoff value of 0.81. The results of detection of *V. cholerae* O1 by RTDs using direct and enrichment methods are highly significant (<0.0001, **Tables 2 and 3**).

We also analyzed the performance of the RDTs using PCR as the gold standard (**Table 4**). In this analysis, the sensitivity and specificity of the RDTs remained between 52.3–58.2% and 92.3–96.8%, respectively. The accuracy was low (>77%) compared to culture techniques as the gold standard. CUI with the positive predictive value was good with SMAR-II and Crystal-VC with the attribute weight ranging from 0.497 to 0.515 and all kits performed well with the negative predictive values with the attribute weights from 0.684 to 0.714. Except for SMART-II kit, comparative detection of *V. cholerae* O1 with PCR assay with the other kits is not significant (**Table 4**). About 15% of the samples gave PCR positivity when both the direct and enrichment culture methods failed to identify *V. cholerae* O1 in the samples and this would have affected the specificity outcome. In these set of samples, RDT results were also negative.

**Table 5** shows Bayesian latent class analysis of RDT results without any gold standard. The sensitivity of the cholera RDT kits in the direct and after enrichment testing of stools showed high in SMART-II (88% and 92%, respectively). The specificity in the direct and after enrichment testing of stools showed high in SD Cholera and Crystal-VC kits (≥ 97%). The overall

**Table 4. Assay results of RDTs and PCR (PCR as gold standard).**

| Assay | | PCR | | | | | | | | | |
|---|---|---|---|---|---|---|---|---|---|---|---|
| | | Positive | Negative | Sens. | Spec. | PPV | NPV | Accuracy | CUI+* | CUI- | p-value# |
| SD Cholera Kit | Positive | 115 | 11 | 52.3% (45.4–59) | 96.1% (93.2–98.1) | 91.3% (84.9–95.6) | 72.4% (67.6–76.8) | 77.1% (73.2–80.7) | 0.477 | 0.695 | 0.5002 |
| | Negative | 105 | 275 | | | | | | | | |
| SMART—II Kit | Positive | 128 | 22 | 58.2% (51.4–64.8) | 92.3% (88.6–95.1) | 85.5% (78.6–90.6) | 74.2% (69.3–78.6) | 77.5% (73.6–81) | 0.497 | 0.684 | 0.0152 |
| | Negative | 92 | 264 | | | | | | | | |
| Crystal VC Kit | Positive | 122 | 9 | 55.4% (48.6–62.1) | 96.8% (94.1–98.5) | 93.1% (87.4–96.8) | 73.8% (69.1–78.2) | 78.8% (75–82.3) | 0.515 | 0.714 | 0.0604 |
| | Negative | 98 | 277 | | | | | | | | |

*CUI; Clinical utility index of the positive and negative test. >0.81 excellent utility; 0.64–0.80 good utility; 0.49–0.63 fair utility; 0.36–0.48; poor utility and <0.36 very poor utility [34].

#Fisher's exact test.

**Table 5. Results of Bayesian-LCM in evaluating the RDTs.**

| Assay | % Sensitivity | 95% CI | % Specificity | 95% CI |
|---|---|---|---|---|
| SD Cholera Dire | 83 | 77–90 | 97 | 96–99 |
| SD Cholera Enrichment | 82 | 76–88 | 100 | 100–100 |
| SMART-II Direct | 88 | 83–94 | 93 | 90–95 |
| SMART-II Enrichment | 92 | 88–97 | 96 | 94–98 |
| Crystal- Direct | 86 | 80–92 | 97 | 95–99 |
| Crystal-VC Enrichment | 91 | 86–96 | 99 | 99–100 |
| Culture Direct | 85 | 79–91 | 97 | 95–99 |
| Culture Enrichment | 95 | 91–98 | 93 | 91–96 |
| PCR *ctx* | 98 | 95–100 | 77 | 73–81 |
| PCR *rfb* | 98 | 95–100 | 77 | 73–81 |

DNA based *ctx* and O1*rfb* PCR assays had a very high sensitivity (98%), but the specificity remained low (77%).

Of the 11 samples with blood in the stools tested in this study, one was positive in the PCR assay. Of the 129 samples that showed positive for *V. cholerae* O1 in direct culture method, 21 samples (16.3%) were also positive for other pathogens (**S1 Table**). The presence of other pathogens did not affect the performance of RDTs or PCR assays. Results of sensitivity/specificity of RDTs and direct culture stratified by age, gender and diarrheal disease severity are shown in **Table 6**. Considering direct culture technique as gold standard, the sensitivity of SMART-II and Crystal VC was 100% and SD Cholera and Crystal VC were more specific (>95%) with the stool samples collected from less than 5 years age group. Among more than 5 years age group, SMART-II was more sensitive (85.2%) and SD Cholera and Crystal VC RDTs are more specific (>93%). We have analyzed RDT results of sensitivity/specificity among male and female patients. SMART-II and SD Cholera are more sensitive and specific, respectively in samples collected from both the genders (**Table 6**). In more sever cases of diarrhea, the sensitivity of all the RDTs remained the same (85%), but Crystal VC was more specific (96.6%) than the rest. Among moderate/mild cases, SMART-II and SD Cholera are more sensitive (86.2) and specific (94.3%), respectively.

Results of sensitivity/specificity of RDTs and PCR assay stratified by age, gender and diarrheal disease severity are shown in **Table 7**. Considering PCR assay as gold standard, the sensitivity of all the RDTs in less than 5 years age group remained low (≤ 38%) and SD Cholera showed more specific (97.7%). However, the sensitivity and specificity are more with SMART-II (60.8) and Crystal VC (97.1%), respectively among more than 5 years age group. SMART-II was more sensitive (56–60%) and Crystal VC are more specific (>95%) in samples collected from both the genders. In more sever cases of diarrhea, the sensitivity of all the RDT

**Table 6. Analysis of sensitivity/specificity of RDTs and direct culture results stratified by age, gender and diarrheal disease severity (culture technique as the gold standard).**

| Assay kit | Age | | | | Gender | | | | Disease severity | | | |
|---|---|---|---|---|---|---|---|---|---|---|---|---|
| | Less than 5 years | | More than 5 years | | Male | | Female | | Severe | | Moderate/mild | |
| | Se | Sp | Se | Sp | Se | Sp | Se | Sp | Se | Sp | Se | Sp |
| SD Cholera | 85.7 | 96.8 | 81.2 | 94 | 83.3 | 92.5 | 79.7 | 96.3 | 85 | 94.9 | 80.7 | 94.3 |
| SMART-II | 100 | 90.3 | 85.2 | 89.5 | 88.3 | 86.1 | 84.1 | 93.1 | 85 | 91.5 | 86.2 | 89.3 |
| Crystal VC | 100 | 95.2 | 82 | 93.3 | 81.7 | 91.4 | 84.1 | 95.8 | 85 | 96.6 | 82.6 | 93.1 |

Sensitivity (Se) and specificity (Sp) values are in percentage. Direct culture technique as gold standard

**Table 7. Analysis of sensitivity/specificity of RDTs and PCR assay results stratified by age, gender and diarrheal disease severity (PCR as the gold standard).**

| Assay kit | Age | | | | Gender | | | | Disease severity | | | |
|---|---|---|---|---|---|---|---|---|---|---|---|---|
| | Less than 5 years | | More than 5 years | | Male | | Female | | Severe | | Moderate/mild | |
| | Se | Sp | Se | Sp | Se | Sp | Se | Sp | Se | Sp | Se | Sp |
| SD Cholera | 26.9 | 97.7 | 55.6 | 95.9 | 53.4 | 93.7 | 51.3 | 98.6 | 52.9 | 95.6 | 52.1 | 96.3 |
| SMART—II | 38.5 | 93.0 | 60.8 | 92.2 | 60.2 | 88.2 | 56.4 | 96.5 | 52.9 | 91.1 | 59.1 | 92.5 |
| Crystal VC | 30.8 | 95.3 | 58.8 | 97.1 | 56.3 | 95.1 | 54.7 | 98.6 | 52.9 | 97.8 | 55.9 | 96.7 |

Sensitivity (Se) and specificity (Sp) values are in percentage. PCR assay as gold standard

kits remained the same (52.9%), but Crystal VC remained comparatively more specific (97.8%). Among moderate/mild cases, SMART-II and Crystal VC/SD Cholera are more sensitive (59.1%) and specific (>96), respectively.

The detection limit of the cholera RDT kits ranged from 6 x $10^7$ CFU (SD-Cholera) to 6 x $10^8$ CFU (SMART-II and Crystal-VC). Cholera RDTs gave positive results up to 14 days when the swabs were seeded with 1 x $10^6$ CFU and stored in Cary-Blair transport medium at ambient temperature. The details on recovery of *V. cholerae* O1 and RDT results during this test period were shown in supplementary **S2 Table**.

## Discussion

Testing of cholera RDTs in a cholera endemic setting like Kolkata has not been made in a systematic approach. Rapid identification of cholera cases will help in making swift responses to control and spread of outbreaks. Information generated from such studies may be useful in understanding the importance of RDT in cholera surveillance and epidemic monitoring and their use as a point of care testing tool in an endemic context. Culture based techniques are routinely used in the laboratory to support the cholera surveillance or even outbreaks. This technique has also been conventionally used as a reference standard for estimating the performance of other tests. However, due to low sensitivity, stool culture underestimate the specificity [20,27]. The general approach for cholera diagnosis and surveillance is based on clinical examination for typical cholera symptoms and culture confirmation of stool specimens if laboratory facilities are available. Poor sampling and transportation delay of stools significantly affects the efficacy of culture methods. Considering these factors, RDTs are recommended as a supplement for cholera surveillance at primary health care level. Presently, RDT is being used during outbreaks and surveillance by the Indian Integrated Disease Surveillance Program in remote areas where the laboratory supports are not adequate.

In the past, several of RDTs have been used during cholera outbreaks/epidemics with varying degree of specificities and sensitivities [28]. Technical note of the Global Task Force for Cholera Control of the World Health Organization provided directions for application of RDTs [27]. The expected minimal performance of RDTs according to this guideline is with the sensitivity of ≥90% and a specificity of ≥85%. With reference to these recommended cutoff values, the present study showed modest sensitivity, but with good specificity of RDT results with direct culture methods as the gold standard. However, the detection accuracy of >90% was seen at least with two kits. In several studies, Crystal-VC has performed with a sensitivity ranging from 58–100% and specificity ranging from 60–100% [27]. The performance of SD Bioline RDT was almost similar to a report from Haiti [29].

In a few investigations, Bayesian-LCM has been assessed to estimate the performance of assays in the absence of a gold standard [30,31]. With the conditional independence between culture, RDT and PCR, the Bayesian-LCM showed slightly better results in determining the

sensitivity and specificity compared to the culture methods as a gold standard [22]. Overall, the sample size in this study was large enough to consider the Bayesian LCM model for the evaluation of the RDT kits [32]. In the absence of a gold standard, the Bayesian-LCM analysis performed with acceptable sensitivity values for SMART-II and Crystal-VC kits and the specificity values for all the RDT kits remained >90%. These results matched a previous study conducted in the Democratic Republic of Congo with Crystal-VC [22]. Since the Bayesian analysis relies on prior hypotheses, we strictly adhered to the protocols and the assays were made under unbiased conditions. Several impeding factors reported in previous studies are considered that includes trained lab personnel, shortening the sample transportation time and processing in the lab, and selective enrolment of the patients in the study.

The low sensitivity of APW enrichment culture might be due to a shorter incubation time given in this study. Generally, the enrichment of stool samples in APW for >6 hrs at 37°C has significantly improved the specificity of the test [33]. The long enrichment step increases the reporting time and hence the rapidity in detecting the cholera is not fulfilled in this instance. Along with culture methods, PCR has also been used in the evaluations of cholera RDTs [34,35]. As observed in this study, low specificity of PCR compared to culture could be due to false negative culture results and/or the result of low yield of stool DNA. False negative culture results may also be encountered due to the presence of low numbers of pathogens, as a consequence of antibiotic use, which in turn influence PCR results [36–38]. With respective to positive for culture and/or PCR and negative to RDTs, quantification of the *V. cholerae* CFU or CT values in quantitative PCR in discordant samples might help to understand this problem. Moreover, PCR cannot be considered as a POC assay as this technique has, thus far, been limited to equipped laboratories due to dependency on complex and good infrastructure, highly skilled manpower and special storage conditions. Overall, the CUI was good to excellent after enrichment RDTs and fair to excellent in direct detection RDTs [39].

It is interesting to note that that the cholera RDT sensitivity values are lower than in evaluations conducted in Africa and Bangladesh settings. Influence of *V. cholerae* O1 numbers in the samples and the performance of assays in different geographical areas is a very interesting but challenging question. The number of *V. cholerae* in stools depends not only on the antibiotic use, but also depend on the time of collection of stools after the onset of the disease, duration of transport, the presence of bacteria phages, etc. This important aspect needs to be addressed in the future studies. Presence of other enteric pathogens along with *V. cholerae* O1 did not show any influence in the performance of RDTs. This aspect was not addressed in other studies. The number of vibrios present in the acute cholera patients may be up to $10^9$ CFU [40] and hence the performances of all the tested cholera RDT kits are essentially good. In stored swab samples in Cary-Blair medium, cholera RDTs gave positive results, confirming that stool swabs could be stored up to two weeks at ambient temperature, which confirms an earlier study [41].

RDTs represent favorable options for the POC diagnosis of cholera during outbreak situations and in resource poor settings due to their simplicity, long shelf life, less cost and detection ability of *V. cholerae* O1 even in the presence of other pathogens in the stools. However, cholera RDTs cannot be considered as a POC tool due to lack of required specificity in different analysis as shown in this study.

## Conclusion

The overall performance of all the three RDT kits is almost similar. Though the PCR assay was superior to the antibody-based RDTs, it cannot be used as a POC tool due the procedural difficulties. The RTDs will add improved value for the clinical management of cholera during outbreaks/epidemics.

## Supporting information

**S1 Table. Number of RDT and PCR positive samples with other pathogens identified along with *V. cholerae* O1.**
(DOCX)

**S2 Table. Duration of *V. cholerae* O1 viability in Cary-Blair medium and RDT results after APW enrichment.**
(DOCX)

## Author Contributions

**Conceptualization:** Bhabatosh Das, Asish K. Mukhopadhyay, Thandavarayan Ramamurthy.

**Data curation:** Goutam Chowdhury, Tarosi Senapati, Puja Bose, Arundhati Deb, Sangita Paul.

**Formal analysis:** Asha Kamath, Debottam Pal, Thandavarayan Ramamurthy.

**Funding acquisition:** Bhabatosh Das, Thandavarayan Ramamurthy.

**Investigation:** Goutam Chowdhury, Bhabatosh Das, Asish K. Mukhopadhyay, Thandavarayan Ramamurthy.

**Methodology:** Goutam Chowdhury, Tarosi Senapati, Bhabatosh Das, Asha Kamath, Debottam Pal, Puja Bose, Arundhati Deb, Sangita Paul.

**Project administration:** Bhabatosh Das, Asish K. Mukhopadhyay, Shanta Dutta, Thandavarayan Ramamurthy.

**Supervision:** Bhabatosh Das, Asish K. Mukhopadhyay, Thandavarayan Ramamurthy.

**Writing – original draft:** Bhabatosh Das, Debottam Pal, Thandavarayan Ramamurthy.

**Writing – review & editing:** Asish K. Mukhopadhyay, Shanta Dutta.

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
