## [Decision Letter · Decision Letter 0]

19 Nov 2020

Dear Dr. T,

Thank you very much for submitting your manuscript "Laboratory evaluation of the rapid diagnostic tests for the detection of Vibrio cholerae O1 using diarrheal samples" for consideration at PLOS Neglected Tropical Diseases. As with all papers reviewed by the journal, your manuscript was reviewed by members of the editorial board and by several independent reviewers. In light of the reviews (below this email), we would like to invite the resubmission of a significantly-revised version that takes into account the reviewers' comments. 

We cannot make any decision about publication until we have seen the revised manuscript and your response to the reviewers' comments. Your revised manuscript is also likely to be sent to reviewers for further evaluation.

Sincerely,

Husain Poonawala

Guest Editor

Mathieu Picardeau

Deputy Editor

Reviewer's Responses to Questions

**Key Review Criteria Required for Acceptance?**

**Methods**

-Are the objectives of the study clearly articulated with a clear testable hypothesis stated?

-Is the study design appropriate to address the stated objectives?

-Is the population clearly described and appropriate for the hypothesis being tested?

-Is the sample size sufficient to ensure adequate power to address the hypothesis being tested?

-Were correct statistical analysis used to support conclusions?

-Are there concerns about ethical or regulatory requirements being met?

Reviewer #1: Overall the methods used are adequate to achive the stated objectives, however the following points could be further clarified. 

1. It is unclear if patients with bloody diarrhea were included as part of the study. This should be clarified. 

2. Authors should clarify if the Crytal VC rapid test detected both O1 and O139.

3. The enriched RDT method should be described in the methods section.

4. A composite outcome using PCR and culture results would be more appropriate to assess the performance of the rapid test. Or otherwise the authors should conduct a separate analysis of the performance of the RDTs using PCR as the gold standard. See these reference: (https://doi.org/10.1111/tmi.13084, https://doi.org/10.1371/journal.pone.0168257 ). 

5. It is unclear what prior distributions were used to inform the pre-test probabilities for culture, PCR and the rapid test. This needs to be clarified. 

6. It will be interesting to provide estimates of the sensitivity/specificity stratified by diseases severity, age and gender. 

7. The sample size calculation is not provided

Reviewer #2: (No Response)

Reviewer #3: yes

**Results**

-Does the analysis presented match the analysis plan?

-Are the results clearly and completely presented?

-Are the figures (Tables, Images) of sufficient quality for clarity?

Reviewer #1: Overall the results follow the analysis plan. The tables and the text support the main conclusion. Some aspects that could be improve include the following points:

1. The analysis of the detection limit seems to indicate a lower quantity of bacteria needed from the SD rapid test to provide a positive result (6x107 vs 6x108). How the authors explain then the lower sensitivity of SD considering these results?

2. The detection limit analysis and the analysis about the duration of positivity from Cary-Blair could be presented in more detail in a supplementary appendix. 

3. It would be good to include a table with the patient’s characteristics.

Reviewer #2: (No Response)

Reviewer #3: Yes

**Conclusions**

-Are the conclusions supported by the data presented?

-Are the limitations of analysis clearly described?

-Do the authors discuss how these data can be helpful to advance our understanding of the topic under study?

-Is public health relevance addressed?

Reviewer #1: Overall the discussion and the conclusions are supported by the results. The main limitations are described. The authors could futher improve the discussion with the following recommendations:

1. The authors should describe in addition to outbreaks how this data is going to help to inform surveillance strategies in endemic places like India. 

2. The low specificity of PCR compare to culture is discussed. This can be the result of false negative culture results as mentioned or as well the result of PCR contamination for example. This is a very important limitation that needs to be further clarify since can widely affect the RDT performance results. 

3. It is interesting to see that sensitivity values are lower than in evaluations conducted in African settings. This could indicate a lower bacterial load in the samples collected in India compared to African setting. Perhaps the authors could discuss this point referring also to other evaluations conducted in Bangladesh (another highly endemic context). 

4. Providing quantification of the CFU when using culture or CT values when using quantitative PRC in different setting might help to understand this issue. Would be possible to provide a hint on this respect, specially for discordant samples (positive for culture and/or PCR and negative to RDTs)?

5. I guess that this lower sensitive will be even more obvious if the PCR is used as a gold standard. The authors could comment on this point as well.

Reviewer #2: (No Response)

Reviewer #3: Yes

**Editorial and Data Presentation Modifications?**

Reviewer #1: (No Response)

Reviewer #2: (No Response)

Reviewer #3: NA

**Summary and General Comments**

Reviewer #1: 16. The authors should describe in addition to outbreaks how this data is going to help to inform surveillance strategies in endemic places like India. 

17. The low specificity of PCR compare to culture is discussed. This can be the result of false negative culture results as mentioned or as well the result of PCR contamination for example. This is a very important limitation that needs to be further clarify since can widely affect the RDT performance results. 

18. It is interesting to see that sensitivity values are lower than in evaluations conducted in African settings. This could indicate a lower bacterial load in the samples collected in India compared to African setting. Perhaps the authors could discuss this point referring also to other evaluations conducted in Bangladesh (another highly endemic context). 

19. Providing quantification of the CFU when using culture or CT values when using quantitative PRC in different setting might help to understand this issue. Would be possible to provide a hint on this respect, specially for discordant samples (positive for culture and/or PCR and negative to RDTs)?

20. I guess that this lower sensitive will be even more obvious if the PCR is used as a gold standard. The authors could comment on this point as well.

Reviewer #2: In this manuscript the authors evaluate the performance of three cholera O1 rapid diagnostic tests. The evaluation is carried out on stool samples of patients with acute diarrhoea admitted to two hospitals in Kolkata. The authors used a “classic” method for evaluation by paring the RDT result with V. choleræ culture result as gold standard. With the knowledge that the culture is an imperfect gold standard, the authors also carried out a second statistical analysis with the use of a Bayesian latent class model. 

The parts of the manuscript related to the methods and the results are straight forward and well explained. There are, however, of major issues that I suggest the authors to address to. These issues are connected to each other and encompass the whole manuscript. I try here to disentangle them.

Major issues:

#1. The authors stated in the introduction (line 111) that “Cholera rapid diagnostic test (RDT) represent a promising tool in the early detection…”. Some RDT (i.e. Crystal VC) are in market for more than 10 years, and in recent years a considerable amount of research has been carried out to understand their pros and cons and the context where RDTs find their deployment. Four years ago, in the light of these works, WHO recommends the use of RDT for early detection and monitoring epidemics, but not for patient’s diagnosis. 

#2. Following the statement above, and considering that the setting of this manuscript (two hospitals in Kolkata) is cholera endemic, it remains unclear the scope in which the authors frame their work on RDTs. As I said, RDTs have a clear role in surveillance and epidemic monitoring and the authors acknowledge that at the beginning of the discussion (Reference 24). In this scope, the work of this manuscript corroborates previous researches. However, if the scope is to evaluate the performance of RDTs in endemic setting, and the main purpose is the patient’s differential diagnosis (as some sentences in the introduction and in the discussion seem to indicate), the authors should clearly state that as the main purpose of their work and develop the discussion in this perspective. 

#3. With the main purpose of the patient’s differential diagnosis in a cholera endemic setting the use of PCR may have a role. However, I would like to point out that PCR, and its evaluation, is not mentioned anywhere in the introduction, while it is extensively mentioned in the “Material and Methods”, “Results” and “Discussion” sections. The authors even concluded the manuscript by acknowledging that “a portable PCR machine along with PCR-dipstick DNA chromatography” (line 385) may be the preferred point of care tool. This last sentence is rather confusing for two reasons: first, because this PCR variant was not evaluated in this work, and second, because the authors do not clarify in which epidemiological context the PCR-dipstick DNA chromatography would be the preferred tool. Moreover the authors should clarify why, after stating that ”PCR cannot be considered as a POC assay” (line 363), and that “PCR … cannot be used as a POC tool due to the procedural difficulties” (lines 380-381), PCR iss eventually considered advantageous as POC test (read lines 383-384 “Considering the performance of PCR, it would be advantageous is adopting the technique for POC test”). 

#4.The authors should also clarify what the purpose of PCR in this work: is it used as reference-standard, as stated in the abstract and in the discussion (lines 351-352), or as a test under evaluation, as described in the results and in other parts of the discussion? The back and forth of PCR as reference-standard and as test under evaluation is a source of confusion for the readers. 

Other minor, but not so minor, issues:

 - Abstract line 42-43: the author mention that RDTs are compared with culture and PCR method. I point out that the comparison is RDTs with PCR as reference was not presented in the manuscript. 

- Abstract lines 54-55: the authors mention here “After enrichment, the high sensitivity…”. At this point of the reading, there is no mention of enrichment procedure, and it not clear in which sample the enrichment was carried out. Only in the main manuscript ("Materials and Methods", "culture technique") it is mentioned that enrichment was carried out to sample before culture. 

- Abstract lines 60-65: The conclusion is confusing as the authors seem to recommend the use of PCR, while few lines before they stated that PCR cannot be used as a point of care tool. See major issue #3 for more comments. 

- Sample collection lines 138-148: I recommend the authors to mention the dates in which the stool samples were taken.

- Data analysis line 222: “The primary endpoint … using stool culture result as the gold standard for comparison”. The authors should clarify here what results among the direct culture of the culture after enrichment is use as gold standard.

- Discussion line 343-344: “The sensitivity of APW enrichment culture was unexpectedly low to compare direct culture method”. One effect of the APW enrichment procedure is to isolate the V. choleræ from other pathogens. So, it is not surprising that less positive sample were found in the culture samples after enrichment when compared with the direct culture where the isolation was not carried out. For this purpose, it would be interesting to compare the performances of two cultures procedures – with and without APW enrichment – with PCR as reference-standard.

Reviewer #3: Review comments uploaded. Accept with minor revision

PLOS authors have the option to publish the peer review history of their article (what does this mean?). If published, this will include your full peer review and any attached files.

Reviewer #1: No

Reviewer #2: No

Reviewer #3: Yes: Munirul Alam
---

## [Decision Letter · Decision Letter 1]

5 Apr 2021

Dear Dr. Ramamurthy,

Thank you very much for submitting your manuscript "Laboratory evaluation of the rapid diagnostic tests for the detection of Vibrio cholerae O1 using diarrheal samples" for consideration at PLOS Neglected Tropical Diseases. As with all papers reviewed by the journal, your manuscript was reviewed by members of the editorial board and by several independent reviewers. In light of the reviews (below this email), we would like to invite the resubmission of a significantly-revised version that takes into account the reviewers' comments. 

The authors are requested to address the comments from reviewer 2 regarding editing the original manuscript and revising the discussion. "Conversely, the authors missed the opportunity to elaborate arguments related to what is specific and innovative in this manuscript, and notably their opinion (or they strategy) in using RDTs as point of care in an endemic contex". A shorter, revised, concise manuscript will be an improvement over the current version and increase the likelihood of publication.

We cannot make any decision about publication until we have seen the revised manuscript and your response to the reviewers' comments. Your revised manuscript is also likely to be sent to reviewers for further evaluation.

Sincerely,

Husain Poonawala

Associate Editor

Mathieu Picardeau

Deputy Editor

The authors are requested to address the comments from reviewer 2 regarding editing the original manuscript and revising the discussion. "Conversely, the authors missed the opportunity to elaborate arguments related to what is specific and innovative in this manuscript, and notably their opinion (or they strategy) in using RDTs as point of care in an endemic contex". A shorter, revised, concise manuscript will be an improvement over the current version and increase the likelihood of publication.

Reviewer's Responses to Questions

**Key Review Criteria Required for Acceptance?**

**Methods**

-Are the objectives of the study clearly articulated with a clear testable hypothesis stated?

-Is the study design appropriate to address the stated objectives?

-Is the population clearly described and appropriate for the hypothesis being tested?

-Is the sample size sufficient to ensure adequate power to address the hypothesis being tested?

-Were correct statistical analysis used to support conclusions?

-Are there concerns about ethical or regulatory requirements being met?

Reviewer #2: (No Response)

**Results**

-Does the analysis presented match the analysis plan?

-Are the results clearly and completely presented?

-Are the figures (Tables, Images) of sufficient quality for clarity?

Reviewer #2: (No Response)

**Conclusions**

-Are the conclusions supported by the data presented?

-Are the limitations of analysis clearly described?

-Do the authors discuss how these data can be helpful to advance our understanding of the topic under study?

-Is public health relevance addressed?

Reviewer #2: (No Response)

**Editorial and Data Presentation Modifications?**

Reviewer #2: (No Response)

**Summary and General Comments**

Reviewer #2: I acknowledge that the authors took in to account most of the suggestions proposed by the reviewers.

However, in may opinion, the discussion, despite its length, merely repeats what already said in the results section (or in the introduction) with some additional references (for example paragraph in lines 414-420 is a repetition of what written in the introduction). Conversely, the authors missed the opportunity to elaborate arguments related to what is specific and innovative in this manuscript, and notably their opinion (or they strategy) in using RDTs as point of care in an endemic context.

PLOS authors have the option to publish the peer review history of their article (what does this mean?). If published, this will include your full peer review and any attached files.

Reviewer #2: No
---

## [Editor Report · Decision Letter 2]

30 May 2021

Dear Dr. Ramamurthy,

We are pleased to inform you that your manuscript 'Laboratory evaluation of the rapid diagnostic tests for the detection of Vibrio cholerae O1 using diarrheal samples' has been provisionally accepted for publication in PLOS Neglected Tropical Diseases.

Best regards,

Husain Poonawala

Associate Editor

Mathieu Picardeau

Deputy Editor

---

## [Editor Report · Acceptance letter]

10 Jun 2021

Dear Dr. Ramamurthy,

We are delighted to inform you that your manuscript, "Laboratory evaluation of the rapid diagnostic tests for the detection of Vibrio cholerae O1 using diarrheal samples," has been formally accepted for publication in PLOS Neglected Tropical Diseases.

Best regards,

Shaden Kamhawi

co-Editor-in-Chief

Paul Brindley

co-Editor-in-Chief
